# The Chalcone Isomerase Family in Cotton: Whole-Genome Bioinformatic and Expression Analyses of the *Gossypium barbadense* L. Response to *Fusarium* Wilt Infection

**DOI:** 10.3390/genes10121006

**Published:** 2019-12-04

**Authors:** Qian-Li Zu, Yan-Ying Qu, Zhi-Yong Ni, Kai Zheng, Qin Chen, Quan-Jia Chen

**Affiliations:** College of Agronomy, Xinjiang Agricultural University, 311 Nongda East Road, Urumqi 830052, China; xjzuqianli@126.com (Q.-L.Z.); xjyyq5322@126.com (Y.-Y.Q.); nizhiyong@126.com (Z.-Y.N.); cqq0777@163.com (Q.C.)

**Keywords:** chalcone isomerase, synteny, qRT-PCR, disease resistance, cotton

## Abstract

Chalcone isomerase (CHI) is a key component of phenylalanine metabolism that can produce a variety of flavonoids. However, little information and no systematic analysis of *CHI* genes is available for cotton. Here, we identified 33 *CHI* genes in the complete genome sequences of four cotton species (*Gossypium arboretum* L., *Gossypium raimondii* L., *Gossypium hirsutum* L., and *Gossypium barbadense* L.). Cotton CHI proteins were classified into two main groups, and whole-genome/segmental and dispersed duplication events were important in *CHI* gene family expansion. qRT-PCR and semiquantitative RT-PCR results suggest that *CHI* genes exhibit temporal and spatial variation and respond to infection with *Fusarium* wilt race 7. A preliminary model of *CHI* gene involvement in cotton evolution was established. Pairwise comparison revealed that seven *CHI* genes showed higher expression in cultivar 06-146 than in cultivar Xinhai 14. Overall, this whole-genome identification unlocks a new approach to the comprehensive functional analysis of the *CHI* gene family, which may be involved in adaptation to plant pathogen stress.

## 1. Introduction

Plants are fixed on land and are constantly challenged by various environmental pressures. In addition, plants exhibit different regulatory responses to a variety of biotic stresses [1]. In the process of evolution, land plants have evolved into a large number of specific metabolites, including a vast variety of flavonoids, to address diverse biotic and abiotic stresses [2]. Flavonoids are a type of secondary metabolite that occur widely in plants [3]. Chalcone isomerase (CHI) is a key component of phenylalanine metabolism that can produce a variety of flavonoids [4]. CHI is the rate-limiting enzyme that plays an initial role in the isoflavone synthesis pathway and catalyzes the production of naringenin [4]. These reactions can occur spontaneously in nature, but CHI can increase the reaction rate by 107 times [5]. Regulating the expression of the *CHI* gene can affect the biosynthesis of various defensive products in the phenylalanine metabolic pathways [6]. At present, there are few reports on the role of *CHI* genes in the response to Fusarium wilt in island cotton. It is crucial to provide good candidate genes for disease resistance breeding of island cotton through identifying the relevant disease resistance genes. Therefore, research on CHI is of great significance.

The CHI protein is found in higher plants as a monomer with approximately 220 residues, and its relative molecular weight is approximately 24,000–29,000 units [7]. The CHI protein homology among different species is relatively high at 49% to 82% in different plants [8]. Two major types of *CHI* genes have been identified in plants. Type I CHI proteins are ubiquitous in vascular plants and catalyze 6-hydroxy-chalcone into (2S)-5-hydroxy-flavonoid. Type II CHI proteins are mostly found in leguminous plants and use 6-hydroxy-chalcone as a substrate for 5-deoxy (iso)-flavonoid [9]. It has also been proposed that the proteinoids in fungi and bacteria can be classified as type III [8].

Accumulating evidence has shown that *CHI* genes play a role in anthocyanin synthesis and defense. The accumulation patterns of the *Ophiorrhiza japonica CHI* gene (*OjCHI*) is consistent across different tissues and developmental stages, indicating its potential role in anthocyanin biosynthesis [10]. Recently, the whole-genome sequencing project of carnation revealed two chalcone isomerase genes (*Dca60979* and *Dca60978*) involved in the coloration of orange flowers [11]. More recent investigations suggested an involvement of *CHI* genes in the abiotic stress responses and biotic stress responses. The expression of *Millettia pinnata CHI* gene (*MpCHI*) in *saccharomyces cerevisiae* salt-sensitive mutants enhances salt-tolerance, so *MpCHI* regulates the resistance of *Saccharomyces cerevisiae* to salt treatment in *Millettia pinnata* [12]. The change of *Deschampsia antarctica CHI* gene (*DaCHI*) expression was regulated by salinity stress, so the *DaCHI* gene may have a unique function of regulating salt stress in the phenylalanine metabolism pathway in *Deschampsia antarctica* [13]. The overexpression of *Glycine max CHI1A* gene (*GmCHI1A*) in soybean hairy roots enhanced daidzein accumulation and conferred resistance to the pathogen, the results demonstrated that overexpression of *GmCHI1A* plays positive roles in the response of soybean to *Phytophthora sojae* [14]. The chalcone isomerase 3 gene (CI3) product induces the biosynthesis of resistant molecules that contribute to resistance to insects, *Fusarium proliferatum*, *Fusarium graminearum,* and *Fusarium verticillioides*, thereby both anti-insect antifungal activity suggests it is valuable for producing higher yielding maize of improved quality [15]. The silencing of the *Gossypium hirsutum CHI* gene (*GhCHI*) results in the loss of resistance to *Verticillium* wilt in cotton using VIGS technology, confirming the function of the *GhCHI* gene in cotton defense against *Verticillium* wilt [16]. The *Gossypium barbadense* CHI (GbCHI) protein inhibits spore germination and mycelial growth of *Verticillium dahlia* [17]. The antioxidant performance of extracts obtained from a parent strain overexpressing the CHI gene was superior to that of the control and parental plant extracts. These results are related to the increased sensitivity of flax hybrids to *Fusarium* infection [18]. Isoflavones, as inhibitors of pathogen infection (phytoalexins) and the CHI gene product, play an important role against pathogens in plant-environment interactions [19]. As an important economical crop, with a worldwide industrial value of approximately 100 billion dollars [20], cotton is an important source of natural fiber for the textile industry [21]. Diploid cotton species such as *G. raimondii* L. (Dt-subgenome, 737.8 Mb) and *G. arboreum* L. (At-subgenome, 1700 Mb) underwent recombination approximately 1–2 million years ago, then underwent polyploidization and gave rise to the allotetraploid cotton species *G. hirsutum* L. (AADD, 2500 Mb) and *G. barbadense* L. (AADD, 2500 Mb) [22]. With the development of sequencing techniques, the genomes of two diploids (*G. arboretum* L. and *G. raimondii* L.) and two allotetraploids (*G. hirsutum* L. and *G. barbadense* L.) have been completely sequenced [23,24,25]. Genome-wide analyses of diverse gene families has successfully provided us with greatly new and important information about the biotic stress response mechanisms of cotton and lay a foundation of critical data for future studies [26].

*CHI* genes have been widely studied in various higher plants, especially with regard to the accumulation and regulation of anthocyanins. However, little information on the cotton *CHI* gene family has been reported, and the involvement of *CHI* genes in disease resistance remains poorly understood. For the first time, we executed a genome-wide survey of *CHI* gene families in four *Gossypium* species. Our previous transcriptomes, which were obtained from the parents and the F_6_ RILs (two resistant offspring and two susceptible offspring) after *Fusarium* wilt infection, have shown that *CHI* genes have an important function in cotton. In our study, 33 *CHI* genes were identified from cotton. We extracted de novo transcriptome sequencing data of *GhCHI* genes in different tissues according to previous studies. Furthermore, we investigated the expression pattern of 10 *GbCHI* genes and *GbCHI* genes in different tissues in response to *Fusarium* wilt infection. In this study, for the first time, phylogenetic tree analysis was used to classify cotton, providing an important breakthrough. This research is also the first to identify and classify the *CHI* gene family in plants by phylogenetic tree analysis and provides novel ideas for future studies of gene families in different species.

## 2. Materials and Methods

### 2.1. Sequence Identification of CHI Proteins

Cotton sequence data (*Gossypium arboretum* L., GRI v1.0; *Gossypium raimondii* L., JGI v2.0; *Gossypium hirsutum* L., ZJU v2.0; *Gossypium barbadense* L., ZJU v1.1) were obtained from the CottonFGD (https://cottonfgd.org/) [27]. Genome data of three other plants (*Arabidopsis thaliana* L.; *Oryza sativa* L.; *Theobroma cacao* L.) were obtained from the Phytozome v12 database (http://phytozome.jgi.doe.gov/pz/portal.html) [28].

The keywords ‘chalcone isomerase’ were used as the query to download the chalcone-like hidden Markov model (HMM) from the Pfam database (https://pfam.xfam.org/) [29]. Initially, HMM profiles of chalcone (PF02431, PF16035, PF16036) were obtained from the local protein BLAST database using HMMER software (http://hmmer.org) [30]. In addition, all redundant sequences were removed manually and further checked for the conserved ‘chalcone’ domain with Pfam 32.0 (http://pfam.xfam.org/) [29] and PROSITE (http://prosite.expasy.org/prosite.html) [31]. The physicochemical properties and subcellular localizations of the CHIs were calculated using the ProtParam tool (https://web.expasy.org/protparam/) and CELLO RESULTS (http://cello.life.nctu.edu.tw/) [32] online. These candidate *CHI* genes were named based on their chromosomal positions (bp) from the At-subgenome chromosome to the Dt-subgenome chromosome.

### 2.2. Phylogenetic, Gene Structure, and Conserved Motif Analyses

To analyze the sequences of CHI proteins of the cotton species and three other plant species, the conserved amino acid sequences of CHI proteins of these species were imported with MEGA7.0 [33], and multiple sequence alignments were performed using ClustalX [34]. The neighbor-joining (NJ) method and bootstrap analysis with 1000 replicates was used to evaluate the significance of nodes, and a phylogenetic tree was constructed with the alignment data [35]. The gene exon–intron structure was illustrated using GSDS 2.0 (http://gsds.cbi.pku.edu.cn/) [36]. Conserved motif analysis of the CHI proteins was performed by MEME (http://meme.sdsc.edu/meme/cgi-bin/meme.cgi) with the minimum and maximum motif widths of six and 50 amino acids, respectively, and the maximum motif number of 15 [37].

### 2.3. Chromosomal Position and Synteny Analysis

Specific chromosomal locations of *CHI* genes were obtained by searching the Phytozome database and the Cotton FGD using default settings. *CHI* genes of four cotton species were displayed separately on the different cotton chromosomes, from the At-subgenome chromosome to the Dt-subgenome chromosome and finally plotted using Mapchart 2.0 [38], based on their physical positions (bp). Synteny analysis of orthologous and paralogous genes across the four cotton species genomes was carried out with the sequences of CHI proteins with MCScanX (http://chibba.pgml.uga.edu/mcscan2/) [39]. Gene duplication events among the four cotton species were elucidated using the viewer package of TBtools [40,41]. To investigate the selective pressure of each homologous *CHI* gene, the value of nonsynonymous substitutions to synonymous substitutions (Ka/Ks) was performed using the PAL2NAL program (http://www.bork.embl.de/pal2nal/) [42].

### 2.4. Analysis of Promoter and Putative MicroRNA Target Site Analysis

The 2000 bp promoter sequences of cotton *CHI* genes were downloaded from Cotton FGD. PLACE (http://www.dna.affrc.go.jp/PLACE/) [43] was used to analyze the *CHI* gene promoter sequences. The psRNATarget database (http://plantgrn.noble.org/psRNATarget/home) [44] was used to predict candidate microRNAs (miRNAs) targeted by *CHI* genes and was searched for complementary regions of the coding sequence (CDS) of *CHI* genes with the default parameters and maximum expectations of 5.0. The candidate microRNA with high degrees of complementarity was selected. 

### 2.5. The Transcriptome of CHI Genes in Cotton

The cotton *CHI* gene FPKMs (fragments per kilobase per million reads) downloaded from ccNET (http://structuralbiology.cau.edu.cn/gossypium/) [45] and Cotton FGD (https://cottonfgd.org/) were used to comprehensively analyze the expression value of cotton *CHI* genes. Finally, heat maps of the cotton *CHI* gene expression values and hierarchical clustering analysis were plotted using Tree View (http://jtreeview.sourceforge.net/).

### 2.6. Plant Materials and Stress Treatments

We surface-disinfected Sea island cotton seeds (Xinhai 14 and 06-146) with 0.5% sodium hypochlorite (NaClO) and sterile distilled water [46]. Then, cotton seeds were transferred to Petri dishes containing filter paper sprinkled with sterile distilled water for two days until they germinated. Seedlings of uniform size and without internal fungal contamination were grown in sterilized soil at 25 °C under a 16/8 h (light/dark) photoperiod for 21 days [46]. *Fusarium* wilt race 7 was used for fungal pathogen inoculation [46] with a spore solution of approximately 10^7^ spores per ml. The 21-day-old seedlings of the susceptible cultivar Xinhai 14 and the resistant cultivar 06-146 were inoculated with *Fusarium* wilt race 7 using the root dip method [46,47]. Seedlings were harvested at 0, 6, 8, 12, 24, and 48 h postinoculation. Moreover, the roots, stems, and leaves of seedlings were harvested at 0 h. After the treatments, harvested seedlings were stored at −80 °C.

### 2.7. Quantitative Real-Time PCR (qRT-PCR) Analysis

Total RNA was extracted using the TRIzol reagent (TianGen, Beijing, China) and treated with RNase-free DNase I (Takara, Dalian, Japan) to remove genomic DNA contamination. First-strand cDNA was synthesized with reverse transcriptase (Takara, Japan). The amplification products were electrophoresed on 2% agarose gel at 100 V in a TAE buffer using 500 bp plus DNA ladder (TianGen, Beijing, China) and 2× PCR Taq MasterMix (Applied Biological Materials, Vancouver, CA, USA). qRT-PCR was performed in an ABI Prism7500 system (Applied Biosystems, Foster City, CA, USA) using SYBR Green Master Mix (Applied Biological Materials, Vancouver, CA, USA) with three replicates for each biological duplicate. The Cotton ubiquitin 7 (*UBQ7*) gene was used as an endogenous control for normalization of template cDNA. The amount of transcript accumulated for cotton *CHI* genes normalized to the internal control was analyzed using the relative 2^−ΔΔCt^ method [48].

## 3. Results

### 3.1. Genome-Wide Survey and Identification of CHI Genes in Four Cotton Species

Sixty-two *CHI* sequences were identified in the cotton using HMMER3.0 to search the local protein BLAST cotton database and Phytozome v12 database. Manual inspection of the results was performed to eliminate different transcripts of the same gene and redundant sequences. The 33 putative CHI proteins were further analyzed by searching the Pfam and PROSITE databases to confirm the presence of the ‘chalcone’ domain. Finally, six gene sequences in *Gossypium arboretum* L., six gene sequences in *Gossypium raimondii* L., ten gene sequences in *Gossypium barbadense* L., eleven gene sequences in *Gossypium hirsutum* L., nine gene sequences in *Arabidopsis thaliana* L., nine gene sequences in *Oryza sativa* L., and eight gene sequences in *Theobroma cacao* L., which were confirmed to contain the “chalcone” domain, were identified. Without proper annotation, the current identifications of these genes were highly confused. For convenience, 33 cotton *CHI* genes were internally assigned a consecutive numbering based on the order of their chromosomal locations (Appendix A). Since *G. raimondii* L. and *G. arboretum* L. were regarded as the At-subgenome and Dt-subgenome ancestors of two allotetraploid cotton species [22,49], the total number of *CHI* genes in the two diploid cotton species were remarkably similar to the total number of *CHI* genes in any one allotetraploid cotton species in our study (Appendix A).

Except for the conserved ‘chalcone’ domain, all cotton CHI proteins varied greatly in size, sequence, and physicochemical properties (Appendix A). The subcellular localizations of the CHI proteins also varied. The average cotton CHI protein length was 314 amino acids, and the lengths varied from 205 (*GbCHI01*; *GbCHI06*; *GhCHI01*; *GhCHI06*) to 449 (*GaCHI04*) amino acids (Appendix A). The isoelectric point (pI) values and molecular weights of the cotton CHI protein sequences ranged from 4.85 to 9.23 and from 22,913.16 to 322,244.76 Da, respectively, according to the ProtParam tool (https://web.expasy.org/protparam/) (Appendix A). Moreover, the predicted subcellular localization of the proteins showed that 15 CHI proteins were located in the chloroplast, while the rest were located in the plasma membrane (7), extracellular tissue (4), cytoplasm (6), and mitochondria (1) (Appendix A).

### 3.2. Phylogenetic Classification, Motif Identification, and Gene Structure

To assess the evolutionary significance of the cotton CHI protein domain structure, we executed a phylogenic analysis according to conserved amino acid sequences among the four cotton species and other species. A large number of branches had higher bootstrap values that statistically indicated similar functions of these homologous proteins with their common ancestors. In general, depending on the substrate, the CHI proteins in plants are classified into type I and type II [9]. The CHI-like proteins in bacteria and fungi or as an enhancer to promote the synthesis of flavonoids are classified as type III and type IV [50,51]. However, based on the phylogenetic trees in this study, CHI proteins could be classified into two major groups, group I and group II (Figure 1), for the first time in cotton presenting a major breakthrough. Group I included 16 members from the cotton species, four members from *Arabidopsis thaliana* L., five members from *Oryza sativa* L., and six members from *Theobroma cacao* L., and Group II included 17 members from the cotton species, three members from *Arabidopsis thaliana* L., three members from *Oryza sativa* L., and four members from *Theobroma cacao* L. (Figure 1). Interestingly, except for the *Thecc1EG045520t1* gene, the cotton CHI proteins were closely related to those of *Arabidopsis thaliana* L., *Oryza sativa* L., and *Theobroma cacao* L. (Figure 1). The CHI proteins of cotton were grouped together based on their branches and the CHI proteins of *Arabidopsis thaliana* (L.), *Oryza sativa* (L.), and *Theobroma cacao* (L.) were also grouped together based on their branches (Figure 1). Additionally, the majority of CHI proteins originating from the At-subgenome were closely grouped together with the CHI proteins of *G. arboreum* L., and the majority of the CHI proteins originating from the Dt-subgenome were closely grouped together with the CHI proteins of *G. raimondii* L. (Figure 1). This result was consistent with the hypothesis that two diploid cotton species underwent recombination and gave rise to allotetraploid cotton species [22,52], which suggests that the *CHI* gene family of allotetraploid cotton species might expand and become more complex according to hybridization and polyploidization events [22]. Compared with *Arabidopsis thaliana* L. and *Oryza sativa* L., CHI proteins in the four cotton species were closely related to those from *Theobroma cacao* L., since they were tightly grouped, as shown in Figure 1.

The ‘chalcone’ domain was approximately 200 amino acid residues in length and was considered to be a crucial element. The CHI motifs were predicted using the program MEME, and six distinct motifs were identified based on the conserved domain alignments of the four cotton species (Figure 2). The frequency of amino acids was expressed by the heights of the letters, and the conservativeness of sequence positions was expressed by the height of the accumulation of all letters (Figure 2). The 41 highly conserved amino acid residues were the same among all the members detected in the ‘chalcone’ repeat regions (Figure 2). The predicted motif positions of CHI proteins, which could be classified into two types by phylogenetic tree analysis, varied among the four cotton species. The conserved domains of group II contained one type of predicted motif position, and group I contained two types of predicted motif positions. For example, the conserved domains of *GaCHI01*, *GhCHI06*, *GhCHI01*, *GrCHI04*, *GbCHI01,* and *GbCHI06* contained the same motif position (Figure 2). The replication and relative positions of motifs in conserved regions of the CHI protein might be decisive for the formation of a specific phenotype.

A survey of the cotton *CHI* gene structure revealed an infrequent distribution of exonic regions (from 4 to 10), indicating crucial evolutionary changes in the cotton genome (Figure 3). The shortest cotton *CHI* genes were *GbCHI01*, *GbCHI06*, *GhCHI01,* and *GhCHI06* with merely 618 bp, whereas the longest one was identified as *CaCHI04* with an 8092 bp genomic sequence (Figure 3). The *CHI* gene clustering among the *CHI* homologous gene pairs in cotton showed highly similar gene structures (Figure 3 and Appendix A). In addition, the gene structures of homologous gene pairs (*GbCHI06*/*GhCHI06*, *GrCHI05*/*GhCHI10*, *GaCHI02*/*GbCHI02,* and *GaCHI03*/*GhCHI03*) were almost identical, with minor differences (Figure 3 and Appendix A).

### 3.3. Chromosomal Distribution and Synteny Analysis of CHI Genes between Gossypium Species

Mapping cotton *CHI* genes onto their chromosomes revealed a nonuniform distribution (Figure 4). The direction of transcription and exact position (in bp) of each cotton *CHI* gene on cotton chromosomes are given in Appendix A. Among them, a total of six *GrCHI* genes, six *GaCHI* genes, 11 *GhGPAT* genes, and 10 *GbCHI* genes were distributed on five chromosomes of *G. raimondii* L., five chromosomes of *G. arboreum* L., 10 chromosomes of *G. hirsutum* L., and nine chromosomes of *G. barbadense* L. chromosomes, respectively (Figure 4). In contrast, each chromosome contained only one gene except for chromosomes A2-chr13, D-chr13, AD2-D13, and AD1-D13, which were confirmed to contain two *CHI* genes (Figure 4). The locations of each cotton *CHI* gene also varied. Interestingly, most cotton *CHI* genes located on different chromosomes appeared at the upper end or lower end of the chromosomes (Figure 4). According to the localization results, there were almost the same numbers of *CHI* genes of At-subgenome and Dt-subgenome in tetraploid cotton, suggesting that the tetraploid cotton *CHI* genes had no obvious preference for the retention and loss of homologous chromosomes between subgenomic groups during the evolution of tetraploid cotton. The *CHI* genes were uniformly distributed on the two homologous chromosomes of At-subgenome and Dt-subgenome tetraploid cotton but were not uniformly distributed on different chromosomes and the distribution was independent of chromosome length.

To illustrate the collinearity relationships of the *CHI* gene family in the four cotton types, the details of the orthologous and paralogous gene pairs are listed in Appendix A and Figure 5. There were 12 orthologous *CHI* gene pairs between *G. hirsutum* L. and the two diploid cotton types, and six pairs were shown for the At-subgenome of *G. hirsutum* L. and *G. arboretum* L. In addition, six pairs were shown between the Dt-subgenome of *G. hirsutum* L. and *G. raimondii* L. (Appendix A and Figure 5). There were 11 pairs orthologous *CHI* genes between *G. barbadense* L. and the two diploid cotton types, and six pairs were shown for the A-genome of *G. barbadense* L. and *G. arboretum* L., and five pairs were shown between the D-genome of *G. barbadense* L. and *G. raimondii* L. (Appendix A and Figure 5). Additionally, there were 11 pairs of orthologous *CHI* genes between the two allotetraploid cotton types (Appendix A and Figure 5). To analyze the gene replication events of the cotton *CHI* gene family, we identified nine paralogous pairs (*GhCHI01/06*, *GhCHI02/07*, *GhCHI03/04*, *GhCHI03/07*, *GhCHI03/08*, *GhCHI10/08*, *GhCHI10/09*, *GhCHI05/10*, and *GhCHI07/08*) of *CHI* genes in the *G. hirsutum* L. genome with the exception of *GhCHI11*, seven paralogous gene pairs (*GbCHI02/07*, *GbCHI03/04*, *GbCHI03/07*, *GbCHI03/08*, *GbCHI04/08*, *GbCHI05/08*, and *GbCHI01/06*) in the *G. barbadense* L. cotton genome, with the exception of *GbCHI10*, and six paralogous gene pairs in the two diploid cotton types (Appendix A and Figure 5). *GaCHI03/04* and *GrCHI01/03* were two paralogous gene pairs that were found in *G. arboretum* L. and *G. raimondii* L., respectively. In the four cotton species, the duplication events of the *CHI* gene family were classified into WGD/segmental, dispersed and proximal duplication, and tandem duplicates, which show other duplicate mechanisms and might have different evolutionary effects, were not discovered (Appendix A and Figure 5). Interestingly, proximal duplication events were found only in *G. raimondii* L. Multiple characteristics of the *CHI* gene family suggested that the main expansion mechanisms in the four cotton species were WGD/segmental and dispersed duplication events rather than proximal and tandem duplication events (Appendix A and Figure 5).

During the evolution of duplicated gene pairs, there are three alternative directions: Non-functionalization, sub-functionalization, and neo-functionalization [53]. To further analyze the adaptive evolution of the CDS regions of cotton *CHI* genes, we calculated the Ka/Ks ratio of each duplicated *CHI* gene pair. In general, a Ka/Ks > 1 indicated that homologous genes underwent positive selection, and a Ka/Ks < 1 indicated purifying selection. In our study, the Ka/Ks ratios of most duplicated gene pairs in the four cotton species did not exceed 1.0, which indicated that they underwent purifying selection (selection against change). However, the Ka/Ks ratios of *GbCHI03*/*GaCHI03*, *GrCHI01*/*GbCHI07*, and *GrCHI01*/*GhCHI08* exceeded 1.0, which suggested positive selection for beneficial mutations (Appendix A). Interestingly, the Ks value of the gene pairs (*GhCHI10*/*GhCHI08*, *GaCHI04/GbCHI07*, *GrCHI04*/*GbCHI06*, and *GrCHI04*/*GhCHI06)* and the Ka value of *GhCHI01*/*GaCHI01* were all zero (Appendix A).

### 3.4. Contained Multiple Stress-Related Cis-Acting Elements and Prediction of MicroRNA Target Sites

By analyzing the promoter of the *CHI* genes, we could predict the function of the *CHI* genes in *G. barbadense* L. The structures of the promoter were identified in *CHI* gene genomic sequences (approximately 2 kb upstream of the presumed initiation codon) using PLACE (Appendix A). MYB and MYC recognition sites in all gene promoters might be responsible for the response to drought and ABA. Notably, CAAT-box and TATA-box recognition sites were the most predominant *cis*-acting elements found in the *CHI* gene promoter regions. Among them, 17 recognition sites were involved in light-responsive elements (I-box, Box4, Sp1, GATA-motif, GT1-motif, GA-motif, TCT-motif, ATCT-motif, chs-CMA1a, MRE, AE-box, AT1-motif, G-Box, ATC-motif, ATC-motif, chs-CMA2a, and LAMP-element), indicating that the *CHI* genes might participate in photosynthesis in *G. barbadense* L. (Appendix A). The MeJA-responsiveness CGTCA-motif (four genes) and TGACG-motif (six genes), gibberellin responsive P-box (seven genes), and TATC-box (one gene), abscisic acid responsive ERE (five genes) and ABRE (nine genes) and salicylic acid responsive TCA-element (seven genes) were also involved in the *CHI* promoter regions (Appendix A). Furthermore, five types of growth and development elements were identified: NON-box (two genes), GCN4-motif (three genes), CAT-box (one gene), O2-site (three genes), and MSA-like (one gene) elements (Appendix A). The MYB binding site was detected in the promoter of three genes and was involved in the regulation of flavonoid biosynthesis-related genes (Appendix A). Collectively, the W-box (seven genes), which are wound-responsive elements, was also contained in the CHI promoter (Appendix A). In addition, a representative *cis*-acting regulatory element involved in anaerobic induction was identified in the promoter of six genes (Appendix A). The data might indicate a major role in the response to various stresses in *G. barbadense* L.

To preliminarily explore the posttranscriptional regulatory mechanisms of miRNA-mediated *CHI* gene families in *G. barbadense* L., we used the psRNATarget server to search the coding sequence regions for possible target sites of miRNAs. The results showed that 239 miRNAs targeted 11 *CHI* genes (Appendix A). For example, *GbCHI05* was targeted by one miRNA, and *GbCHI01* was targeted mostly by 72 miRNAs (Appendix A).

### 3.5. Expression Analysis of CHI Genes in Cotton

We investigated the previously published transcriptome profile across various tissue types to determine the critical role of *CHI* genes in *G. barbadense* L. organ development (Appendix A and Figure 6). We also investigated the expression levels of *CHI* genes in different tissues of *G. hirsutum* L. (Xin Hai14 and 06-146) with gene-specific primers (Appendix A). Based on the heat map, we found that *GhCHI01*, *GhCHI05*, *GhCHI06,* and *GhCHI10* were specifically expressed in the leaf, pistil, and torus and that *GhCHI09* was mainly expressed in the torus and not expressed in the stem (Appendix A and Figure 6). The 10 *CHI* genes could be clustered into groups with two patterns in Xinhai 14 and 06-146, respectively (Figure 6). As shown in Figure 6, the first group in Xinhai 14, including *GbCHI02*, *GhCHI03*, *GhCHI07*, *GhCHI08,* and *GhCHI10*, might execute some critical roles in roots. In the second group in Xinhai 14, the expression levels of five *GbCHI* genes in leaves and stems were higher than those in roots (Figure 6). In the cultivar 06-146, we also observed that the expression levels of *GbCHI04*, *GhCHI07*, *GhCHI08,* and *GhCHI10* in roots were relatively lower, while six genes were highly expressed in roots in another group (Figure 6).

### 3.6. Transcription Levels of GbCHI Genes under Fusarium Wilt Treatment

The expression levels of genes can further provide important information regarding gene function. To decipher the role of *G. barbadense* L. *CHI* genes in the biotic stress response, the expression levels of 10 candidate genes were analyzed after inoculation in Xinhai 14 and 06-146 Sea island cotton cultivars with *Fusarium* wilt race 7 at 0, 4, 8, 12, 24, and 48 h postinoculation using qRT-PCR and semiquantitative RT-PCR analysis (Figure 7 and Appendix A). As previously reported, the *GhCHI* gene played a vital role in the resistance to *Verticillium* wilt, which has strong pathogenicity in *G. hirsutum* L. [16]. The gene-specific primers are listed in Appendix A. qRT-PCR analysis indicated that 10 genes of Xinhai 14 and 06-146 displayed variations in expression patterns in response to biotic stress over the course of the experiments (Figure 7 and Appendix A). Xinhai 14 and 06-146 exhibited susceptibility and resistance to *Fusarium* wilt race 7, respectively [47]. The results indicated that these *CHI* genes were meaningfully induced and rapidly reached peak expression levels under the *Fusarium* wilt race 7 challenge at different time points (Figure 7 and Appendix A). For the overall observation, a total of five cotton *CHI* genes, including *GbCHI01*, *GbCHI05*, *GbCHI06*, *GbCHI09*, and *GbCHI10*, showed different degrees of upregulation (> 2-fold) and expression after *Fusarium* wilt race 7 inoculation in both the susceptible cultivar Xinhai 14 and the resistant cultivar 06-146 (Figure 7). As shown in Figure 7 and Appendix A, two *CHI* genes (*GbCHI04* and *GbCHI08*) showed upregulated (> 2-fold) expression in the resistant cultivar 06-146, while showing no significant change in the susceptible cultivar Xinhai 14. *GbCHI01* and *GbCHI06* showed the highest overall expression levels at the 24 and 48 h and over a 12-fold increase between 0 and 24 h in cultivar 06-146 (Figure 7). Interestingly, *GbCHI10* showed the highest overall expression levels at the 8 h, with over 1-fold the expression as that in Xinhai 14 (Figure 7). A pairwise comparison between *CHI* gene expression patterns of cultivar Xinhai 14 versus cultivar 06-146 under the six time points of the inoculation treatment revealed that seven *CHI* genes (*GbCHI01*, *GbCHI04*, *GbCHI05*, *GbCHI06*, *GbCHI08*, *GbCHI09*, and *GbCHI10*) of cultivar 06-146 showed higher expression levels than those in cultivar Xinhai 14 (Figure 7 and Appendix A). In addition, these results demonstrated that the expression patterns of three *CHI* genes (*GbCHI02*, *GbCHI03*, and *GbCHI07*) in cultivar 06-146 were nearly the same as those in cultivar Xinhai 14 (Appendix A).

## 4. Discussion

In this study, 33 *CHI* genes were identified in cotton, and we also searched *CHI* genes in three other plant genomes (*Arabidopsis thaliana* L., *Oryza sativa* L., and *Theobroma cacao* L.), which are representative plants of their evolutionary nodes. The distribution of *CHI* genes on chromosomes showed a certain physical position, and the number of *CHI* genes on the chromosomes was relatively low, but the gene expression was not substantially influenced by the physical regions of the *CHI* genes. In a recent study, despite the relocation of the chromosomal regions, there was no significant deviation in the proportion of genes that changed expression on any of the chromosomes [54]. The diversity of exon-intron structures also plays a key role in evolution, and the loss of exons–introns might result from fusion and rearrangement of disparate chromosome segments in numerous gene families [55]. In our study, it was found that the exon number of 11 genes was as high as 10, which is very rare among gene structures, and *CHI* genes are generally composed of three or four exons [56]. In Figure 2 and Figure 3, the *CHI* genes in the same group have highly similar exon numbers. For example, *GrCHI03* and *GhCHI07* share the same branch in the phylogenetic tree and show similar exon number and structure. The gene structure of *CHI* genes can reflect the types of predicted motif positions among the *CHI* genes. There were differences in the gene structure of *CHI* genes among different genes and species, which may lead to different amino acid sequences encoded by *CHI* genes and different catalytic activities.

It has been reported that genome-wide replication might be considered a new evolutionary approach that could serve as a source of adaptive phenotypes [57]. Then, we might consider that such genes encoding the products of specific interactions between biological and abiotic exogenous factors are more likely to be preserved in structure and function after replication [57]. Polyploidization is a basal pattern of speciation in plant evolution, leading to the formation of a number of duplicated genes in plant genomes [58] and causing various changes in the expression of genes and organization of genomes [59], and polyploidization provides basic material for speciation, morphological innovation, and adaptation. Gene duplication events play a key role in the expansion of gene families. Since the genomes of *G. raimondii* L. and *G. arboreum* L. have undergone at least two WGD events [30], all tetraploid cotton species are formed by the natural hybridization of *G. raimondii* L. and *G. arboretum* L. [60]. Therefore, the evolution of the four cotton species is closely related. From an evolutionary point of view, each *GaCHI* gene corresponds to two orthologous genes in *G. hirsutum* L. and *G. barbadense* L. and to one orthologous gene in *G. raimondii* L., since the genomes of *G. raimondii* L. and *G. arboreum* L. underwent at least two WGDs [22]. In Figure 1, orthologous *CHI* gene pairs clustered to the same group or the same branch. The results of phylogenetic and collinearity analyses show that the gene numbers, synteny of orthologous *CHI* gene pairs, and phylogeny of *CHI* genes were basically consistent with the evolutionary point of view (Figure 1 and Figure 5, Appendix A). Moreover, the results indicate that WGD/segmental and dispersed duplication events affected the *CHI* gene family expansion in the four types of cotton (Appendix A). The Ka/Ks ratios of approximately 62.5% of gene pairs were less than 0.5, and approximately 4.5% were greater than 1.0 in our study, which meant almost all gene pairs underwent purifying selection (Appendix A); several relevant clues were provided, such as phylogenetics, collinearity, type of expansion, and Ka/Ks values (Figure 1 and Figure 5, Appendix A). Then, a preliminary model of the *CHI* gene family loci involvement in the evolutionary process of *Gossypium* was established (Figure 8). In the model, the *CHI* genes descended from the At-subgenome and Dt-subgenome, which evolved into *G. arboreum* L. and *G. raimondii* L., respectively. The *CHI* gene families of the At-subgenome and Dt-subgenome also fused and formed expanding gene families in neoallopolyploids, according to the hybridization, polyploidization, and gene loss events approximately 1–2 Mya (Figure 8). In Figure 8, we found that *GaCHI06* might have been lost in the evolution of *Gossypium hirsutum* L. and that GaCHI06 and GrCHI03 might have been lost in the evolution of *Gossypium barbadense* L. Among the *GbCHI* genes, *GbCHI01/02/03/04/05* originated from *GaCHI01/02/03/04/05*, and *GbCHI06/07/08/09/10* originated from *GrCHI04/01/02/05/06* (Figure 8). Additionally, *GhCHI01/02/03/04/05* originated from *GaCHI01/02/03/04/05*, and *GhCHI06/07/08/09/10/11* originated from *GrCHI04/03/01/02/05/06* (Figure 8).

In recent years, miRNA-mediated gene expression has received widespread attention [61]. During the stage of pathogenic infection, most of the identified miRNAs, such as miR156, miR169, miR171, miR172, miR399, miR477, miR535, and miR827, were significantly expressed in cotton plants [61]. Nevertheless, the majority of miRNAs that targeted *CHI* genes were related to cotton immunity (Appendix A), and the same families of miRNAs were observed between our results and previous studies [61]. This suggests that the CHI protein family has different functions during the progression of cotton disease resistance. The absence or presence of *cis*-elements in promoters may have an important impact on the interpretation of divergences in gene functions. For example, as a plant hormone, MeJA plays a key role in biological stress signaling and helps to improve the disease resistance of *Panax notoginseng* cultivars [62]. Here, our data indicated that six *GbCHI* genes were upregulated under inoculation treatment, which all contained MeJA-responsive *cis*-acting regulatory elements in their promoters (Appendix A).

It has been reported that different *CHI* genes are expressed differently and play different physiological roles in various tissues. For instance, the expression of the *CHI* gene was highest in flowers and lowest in the stems of *Fagopyrum dibotrys* [63]. In addition, overexpression of the *CHI* gene in hairy roots was used to improve the levels of flavonoids [64]. Previous studies have shown that plant flavonoids, including phytoalexins, allelochemicals, and signaling molecules, have critical functions in interactions between microbial pathogens and plants [65] and play important roles in improving plant disease resistance [66]. *CHI* genes are absolutely necessary in the flavonoid biosynthesis pathway [8]. *CHI* genes play key roles in cotton disease resistance; for instance, the *GbCHI* protein inhibits the germination of spores and mycelial growth of *Verticillium dahliae* [16]. From an applied perspective, the high expression level of cotton *CHI* genes might potentially improve resistance to pathogenic bacteria.

In this study, qRT-PCR and semiquantitative RT-PCR expression patterns of each *GbCHI* gene under inoculation treatment provided significant arguments for functional analysis (Figure 7 and Appendix A). In our study, all five *GbCHI* genes were upregulated (> 2-fold) in cultivar Xinhai 14 and cultivar 06-146 (Figure 7). As shown in Figure 7, those *CHI* genes might be involved in the defense against *Fusarium* Wilt. Former studies had shown that preformed defense responses include thickening of the cuticle, synthesis of phenolic compounds, changes in the cell wall structure, accumulation of reactive oxygen species, release of phytoalexins and the hypersensitive response [67], and the responding mechanism between gene expression and pathogen defense needs to be further explored. Pairwise comparison of *CHI* genes between cultivar Xinhai 14 and cultivar 06-146 suggested that *CHI* genes might be involved in resistance to *Fusarium* wilt race 7, and the CHI-mediated regulation of the phenylalanine metabolism programs might be related to the difference in resistance to *Fusarium* wilt race 7 between Xinhai 14 and 06-146 (Figure 7 and Appendix A). These expression variations implied that the cotton *CHI* genes mainly regulate a complex signaling pathway to carry out different physiological functions for defense against pathogenic infection. For instance, transcripts of *GbCHI01*, *GbCHI05*, *GbCHI06*, *GbCHI09,* and *GbCHI10* were abundantly induced by inoculation treatments and made a contribution mainly to the biotic stress response preliminarily (Figure 7). Collectively, the expression levels of the orthologous genes *GbCHI01/06* and *GbCHI05/09* were the same in *G. barbadense*. Therefore, our results indicated that the functions of these two orthologous *CHI* gene pairs did not change significantly with subsequent evolution, and purifying selection might be of great help in maintaining the functions of these two homologous *CHI* gene pairs (Figure 7, Appendix A and Appendix A).

## 5. Conclusions

In summary, we identified 33 *CHI* genes based on the conserved ‘chalcone’ domain that were divided into two main groups in cotton. Subsequently, a preliminary model of *CHI* gene family loci involvement in the evolutionary process of cotton was established. Moreover, the qRT-PCR and semiquantitative RT-PCR expression patterns suggest that the *CHI* gene family had temporal and spatial characteristics triggered by the *Fusarium* wilt race 7 infection. Overall, our analysis of the cotton *CHI* gene family broadened our understanding of the role of *CHI* genes following pathogen inoculation, providing a basis for further understanding the functions of the *GbCHI* gene family and for potential applications in cotton genetic improvement.

## Figures and Tables

**Figure 1 genes-10-01006-f001:**
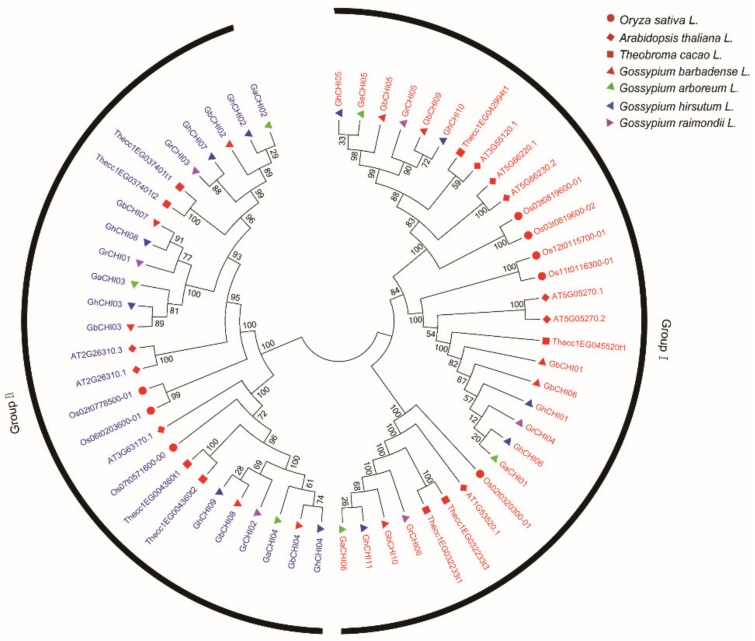
Phylogenetic relationships of cotton CHI proteins. The conserved amino acid sequences of CHI proteins of cotton and three other species were constructed with MEGA7.0. A phylogenetic tree was constructed by the neighbor-joining (NJ) method and bootstrap analysis with 1000 replicates. The bootstrap values are shown at the nodes.

**Figure 2 genes-10-01006-f002:**
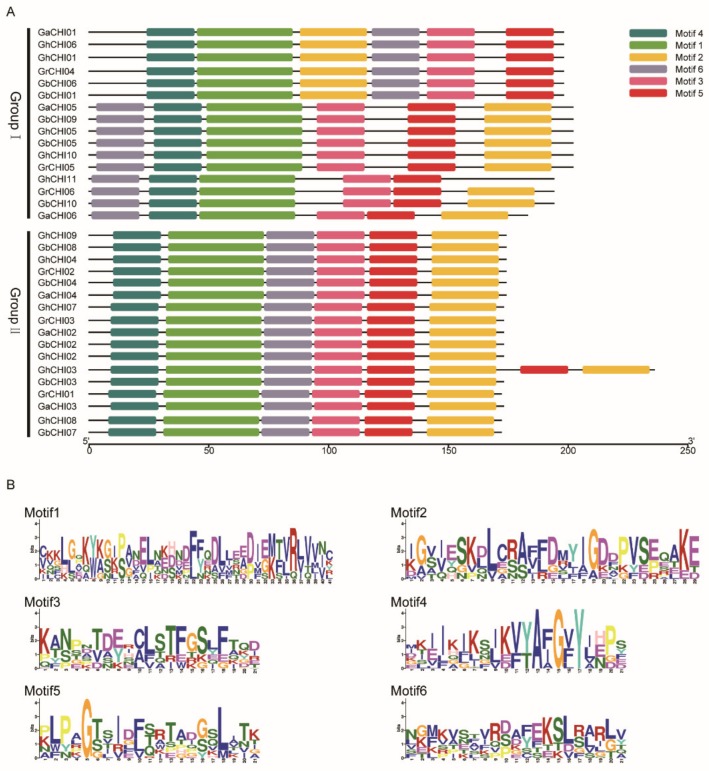
Sequence conservation analysis of CHI proteins. (**A**) The sequence logos of CHI proteins were based on alignments of all the conserved domains in cotton. (**B**) Six conserved motifs were provided.

**Figure 3 genes-10-01006-f003:**
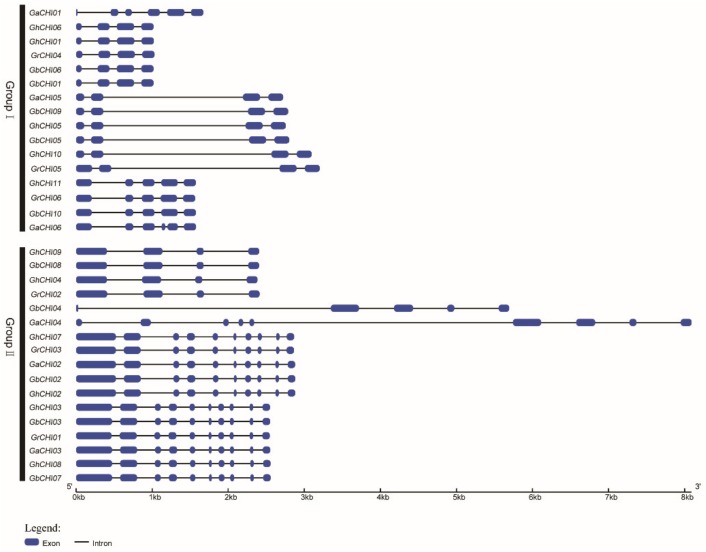
Genomic exon–intron structures of *CHI* genes in cotton. Blue indicates exons; black indicates introns.

**Figure 4 genes-10-01006-f004:**
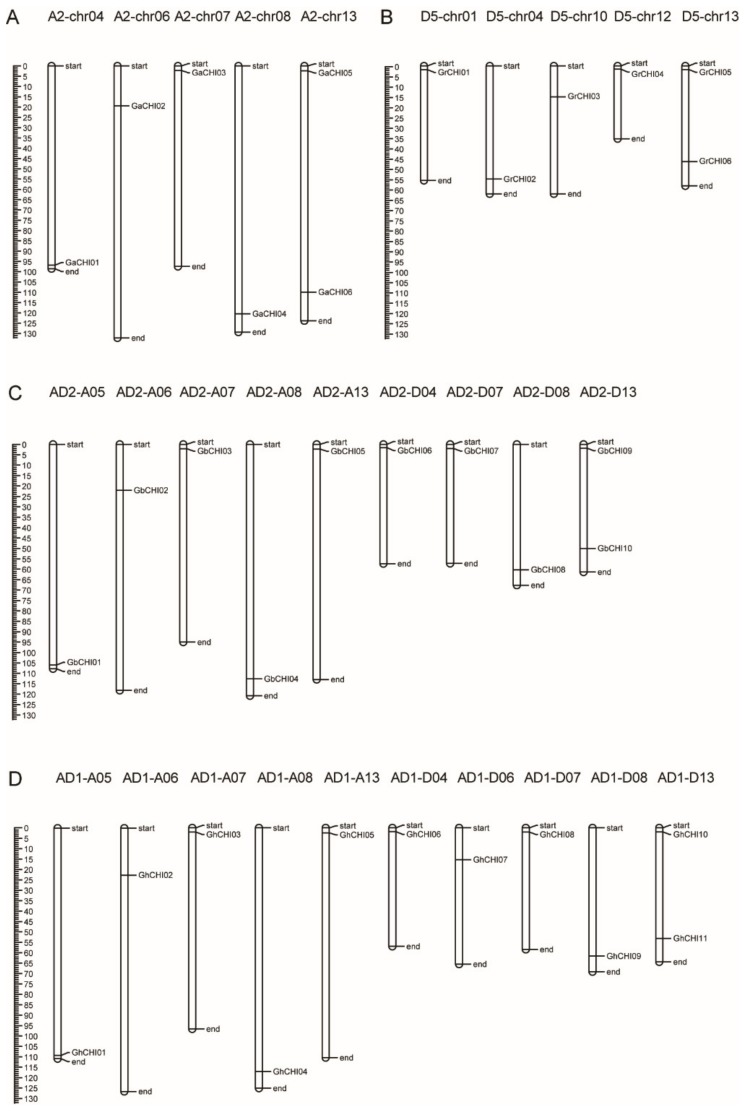
Distribution of 33 *CHI* genes on cotton chromosomes. (**A**) *Gossypium arboretum* L.; (**B**) *Gossypium raimondii* L.; (**C**) *Gossypium barbadense* L.; (**D**) *Gossypium hirsutum* L.; chromosomal distances are given in Mb.

**Figure 5 genes-10-01006-f005:**
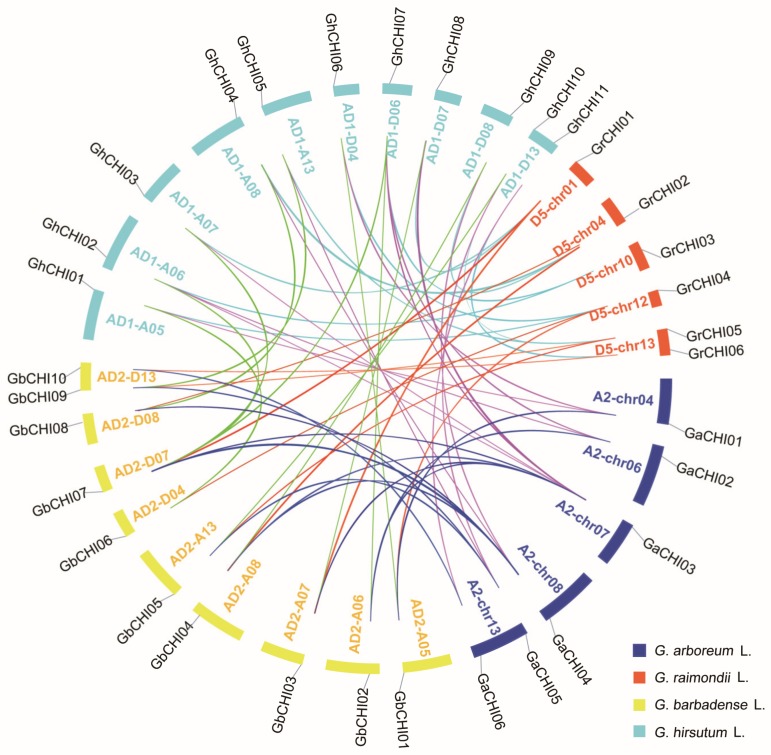
Comparative analysis of synteny and expansion of *CHI* genes in *Gossypium.* A synteny examination of orthologous and paralogous genes performed by MCScanX. Gene duplication events were exhibited using the viewer package of TBtools.

**Figure 6 genes-10-01006-f006:**
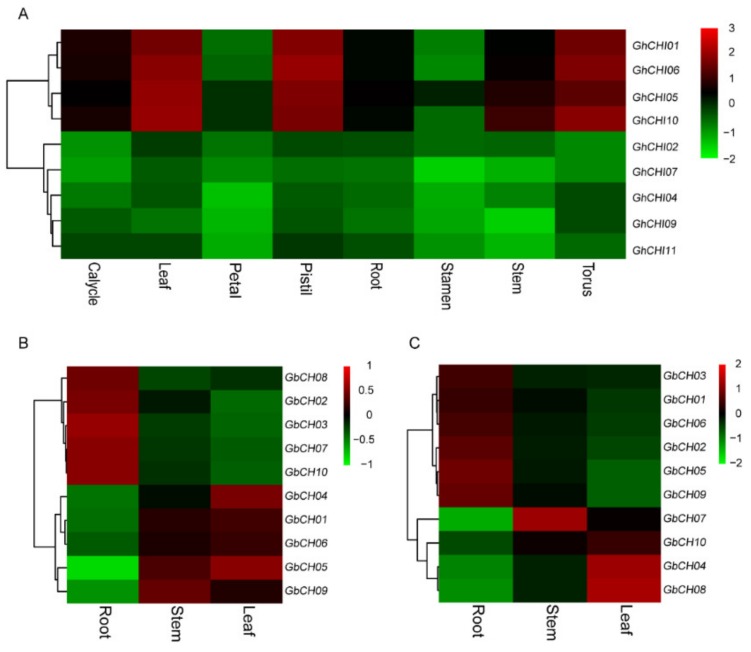
Transcriptional profiling and expression levels of *GhCHI* genes and *GbCHI* genes in different tissues. (**A**) The qRT-PCR expression patterns of the *CHI* genes in *Gossypium hirsutum* L. are shown as the log2 value of fragments per kilobase per million reads (FPKMs). (**B**) The qRT-PCR expression patterns of the *CHI* genes in cultivar Xinhai 14. (**C**) The qRT-PCR expression patterns of the *CHI* genes in cultivar 06-146.

**Figure 7 genes-10-01006-f007:**
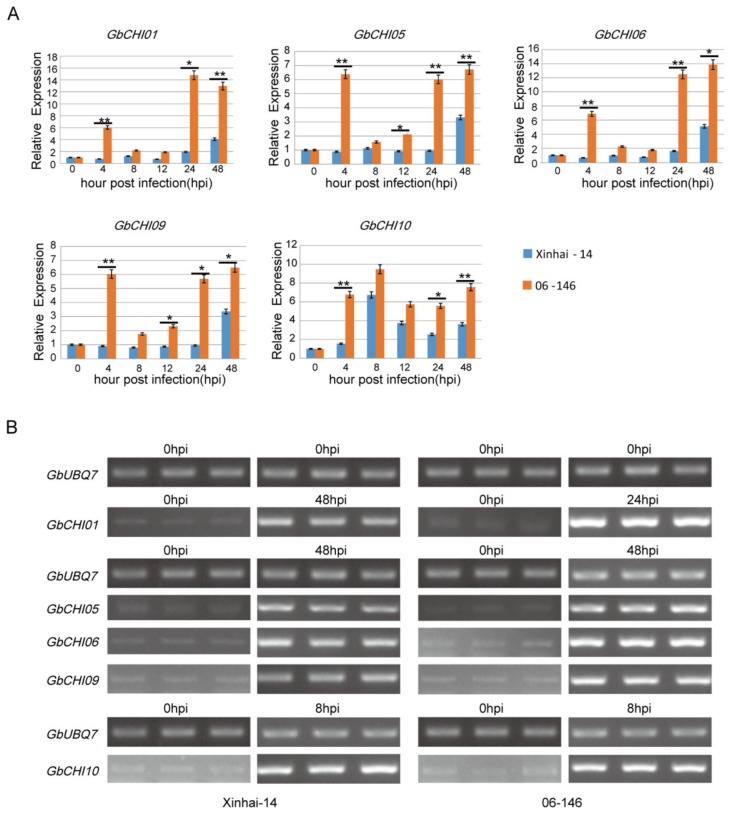
Expression pattern of ten *GbCHI* genes in Xinhai 14 and 06-146. *GbUBQ7* was used as an internal reference to normalize the expression data. (**A**) Expression pattern based on qRT-PCR. Error bars indicate standard errors of three biological replicates. Statistically significant differences between Xinhai-14 and 06-146 were calculated based on an independent Student’s t-test: * *p* < 0.05; ** *p* < 0.01. (**B**) Expression pattern based on RT-PCR gel. 0 hpi (0 h post infection), 8 hpi (8 h post infection), 24 hpi (24 h post infection), and 48 hpi (48 h post infection).

**Figure 8 genes-10-01006-f008:**
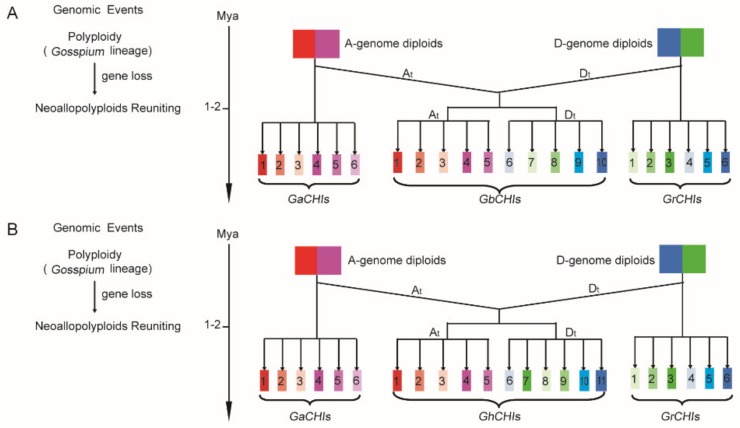
A preliminary model of *CHI* gene family loci involvement in the evolutionary process of cotton. (**A**) A preliminary evolutionary model of the *GbCHI* gene family. (**B**) A preliminary evolutionary model of the *GhCHI* gene family.

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
