# Peer review of "The Chalcone Isomerase Family in Cotton: Whole-Genome Bioinformatic and Expression Analyses of the Gossypium barbadense L. Response to Fusarium Wilt Infection"

_genes, 2019, doi:10.3390/genes10121006_

Round 1

Reviewer 1 Report

Please find below my comments on the paper submitted to “Genes”. Manuscript ID: genes-642085. Title: The Chalcone Isomerase Family in Cotton: Whole-Genome Bioinformatic and Expression Analyses of the Gossypium barbadense L. Response to Fusarium Wilt Infection. Authors: qianli zu, Yan-Ying Qu, Zhiyong Ni, kai zheng, qin chen, Quanjia Chen.

The authors present a comprehensive genetic analysis of the Chalcone Isomerase gene family in Cotton. 33 CHI genes in the genomes of four cotton species (Gossypium arboretum L., Gossypium raimondii L., Gossypium hirsutum L. and Gossypium barbadense L.) have been identfied. qRT-PCR data is presented suggesting that the expression of CHI cotton genes diverges temporally and spatially as well as respond to infection to the pathogen Fusarium wilt race 7. The authors use available transcriptome profile data to analyse the expression of these genes in organ development and tissue. They propose a preliminary model for the involvement of the cotton CHI gene in evolution and predict its involved in adaptation to plant pathogen stress.

Comments

The introduction needs major re-writing. It feels like an amalgam of information not very well connected nor properly explained. As an example pelase see points (1 to 6).

1-“The accumulation patterns of the OjCHI gene are consistent across different tissues and developmental stages, indicating the potential role of OjCHI in anthocyanin biosynthesis in Ophiorrhiza japonica”- comment- define the species of OjCHI first. Like “The accumulation patterns of the Ophiorrhiza japonica CHI gene (OjCHI) is consistent across different tissues and developmental stages, indicating its potential role anthocyanin biosynthesis”.

2-Same for “MpCHI”

3- I do not understand the statement “MpCHI regulates the resistance of Saccharomyces cerevisiae to salt treatment in Millettia pinnata [12].” Comment-What does budding yeast has to do with this? Please explain.

4- “Salinity stress resulted in changes in the expression of the DaCHI gene, so the DaCHI2 gene may have a unique function of regulating salt stress in the phenylalanine metabolism pathway in Deschampsia antarctica [13].” Comment -Re-phrase bad English. Also, one time you call the gene DaCHI and then is DaCHI2?

5- “The results demonstrated that overexpression of GmCHI1A plays positive roles in the response of soybean to Phytophthora sojae [14].” Comment-Which results?

6- “The chalcone isomerase 3 gene (CI3) product … thereby promoting sustainable production and providing better quality food [15].” Comment- I don’t understand the phrase, why does this promote sustainable production and better quality food? Please explain better.

7- “chalcone (PF02431), chalcone (PF16035) and chalcone (PF16036)” Comment- Change to “chalcone (PF02432, PF16035, PF16036)”

8- Table S1. Comment- Change (table S1) column from “subcellular localization" to "predicted subcellular localization".

9- Supplementary table order. Comment- Reformat the numbering of the supplementary tables so that they appear in ascending order in the text and not randon (example, Table S7 is mentioned first in the text than Table S3).

10- Table S5- Comment- As it is this table is totally redundant with the text. Instead of the numbers of genes containing those elements please write  the accession number of the genes that have each of the elements.

11- Table S6- Comment- What is the comparison base line in this data, what are these numbers referenced to? Please specify in a clear way.

12- Figure 7- Comment- This figure should be made bigger and only the genes that show different expression between the two cultivars should be shown. The others move to supplementary data.

13- Figure 7-Please provide RT-PCR gel showing the induction of expression for the most interesting genes and at the most interesting data-points

Author Response

Dear Editors and Reviewers:

    Thank you for your letter and for the reviewers’ comments concerning our manuscript entitled “The Chalcone Isomerase Family in Cotton: Whole-Genome Bioinformatic and Expression Analyses of the Gossypium barbadense L. Response to Fusarium Wilt Infection” (ID: genes-642085). Those comments are all valuable and very helpful for revising and improving our paper, as well as the important guiding significance to our researches. We have studied comments carefully and have made correction which we hope meet with approval. And our manuscript have underwent extensive English editing by American Journal Experts on November 1, 2019. Revised portion are marked in red in the paper. The main corrections in the paper and the responds to the reviewer’s comments are flowing:

Response to Reviewer 1 Comments

Point 1: “The accumulation patterns of the OjCHI gene are consistent across different tissues and developmental stages, indicating the potential role of OjCHI in anthocyanin biosynthesis in Ophiorrhiza japonica”- comment- define the species of OjCHI first. Like “The accumulation patterns of the Ophiorrhiza japonica CHI gene (OjCHI) is consistent across different tissues and developmental stages, indicating its potential role anthocyanin biosynthesis”.

Response 1: We are very sorry for our incorrect writing. We have re-written this part according to the Reviewer’s suggestion. The text of “The accumulation patterns of the OjCHI gene are consistent across different tissues and developmental stages, indicating the potential role of OjCHI in anthocyanin biosynthesis in Ophiorrhiza japonica” were corrected as “The accumulation patterns of the Ophiorrhiza japonica CHI gene (OjCHI) is consistent across different tissues and developmental stages, indicating its potential role anthocyanin biosynthesis”.  

Point 2: Same for “MpCHI

Response 2: We are very sorry for our incorrect writing. We have re-written this part according to the Reviewer’s suggestion. The text of “MpCHI” were corrected as “Millettia pinnata CHI gene”.

Point 3: I do not understand the statement “MpCHI regulates the resistance of Saccharomyces cerevisiae to salt treatment in Millettia pinnata [12].” Comment-What does budding yeast has to do with this? Please explain.

Response 3: We are very sorry for our unclear writing. We have re-written this part according to the Reviewer’s suggestion. Salt-sensitive yeast transformed by the plasmid pYEST2-MpCHI increased in salt tolerance, indicating the possibility that the MpCHI may be involved in M. pinnata salt tolerance in a direct manner. So the text of “MpCHI regulates the resistance of Saccharomyces cerevisiae to salt treatment in Millettia pinnata” were rephrased as “More recent investigations suggested an involvement of CHI genes in the abiotic stress responses and biotic stress responses. The expression of Millettia pinnata CHI gene (MpCHI) in saccharomyces cerevisiae salt-sensitive mutants enhances salt-tolerance, so MpCHI regulates the resistance of Saccharomyces cerevisiae to salt treatment in Millettia pinnata”.

Point 4: “Salinity stress resulted in changes in the expression of the DaCHI gene, so the DaCHI2 gene may have a unique function of regulating salt stress in the phenylalanine metabolism pathway in Deschampsia antarctica [13].” Comment -Re-phrase bad English. Also, one time you call the gene DaCHI and then is DaCHI2?

Response 4: We are very sorry for our bad english. We have re-written this part according to the Reviewer’s suggestion. The text of “Salinity stress resulted in changes in the expression of the DaCHI gene, so the DaCHI2 gene may have a unique function of regulating salt stress in the phenylalanine metabolism pathway in Deschampsia antarctica” were rephrased as “The change of Deschampsia antarctica CHI gene (DaCHI) expression was regulated by Salinity stress, so DaCHI gene may have a unique function of regulating salt stress in the phenylalanine metabolism pathway in Deschampsia antarctica”.

Point 5: “The results demonstrated that overexpression of GmCHI1A plays positive roles in the response of soybean to Phytophthora sojae [14].” Comment-Which results?

Response 5: We are very sorry for our unclear  writing. We have re-written this part according to the Reviewer’s suggestion. The results is “The overexpression of Glycine max CHI1A in soybean hairy roots enhanced daidzein accumulation and conferred resistance to the pathogen”. So the text of “The results demonstrated that overexpression of GmCHI1A plays positive roles in the response of soybean to Phytophthora sojae” were rephrased as “The overexpression of Glycine max CHI1A gene (GmCHI1A) in soybean hairy roots enhanced daidzein accumulation and conferred resistance to the pathogen, the results demonstrated that overexpression of GmCHI1A plays positive roles in the response of soybean to Phytophthora sojae”.

Point 6: “The chalcone isomerase 3 gene (CI3) product … thereby promoting sustainable production and providing better quality food [15].” Comment- I don’t understand the phrase, why does this promote sustainable production and better quality food? Please explain better.

Response 6: We are very sorry for our unclear  writing. We have re-written this part according to the Reviewer’s suggestion. The text of “The chalcone isomerase 3 gene (CI3) product induces the biosynthesis of resistant molecules that contribute to resistance to insects, Fusarium proliferatum, Fusarium graminearum and Fusarium verticillioides, thereby promoting sustainable production and providing better quality food” were rephrased as “The chalcone isomerase 3 gene (CI3) product induces the biosynthesis of resistant molecules that contribute to resistance to insects, Fusarium proliferatum, Fusarium graminearum and Fusarium verticillioides, thereby both anti-insect antifungal activity suggests it is valuable for producing higher yielding maize of improved quality”.

Point 7: “chalcone (PF02431), chalcone (PF16035) and chalcone (PF16036)” Comment- Change to “chalcone (PF02432, PF16035, PF16036)”

Response 7: We are very sorry for our incorrect writing. We have re-written this part according to the Reviewer’s suggestion. The text of “chalcone (PF02431), chalcone (PF16035) and chalcone (PF16036)” were corrected as “chalcone (PF02432, PF16035, PF16036)”.

Point 8: Table S1. Comment- Change (table S1) column from “subcellular localization" to "predicted subcellular localization".

Response 8: We are very sorry for our incorrect writing. We have re-written this part according to the Reviewer’s suggestion. The text of “subcellular localization” were corrected as “predicted subcellular localization”.

Point 9: Supplementary table order. Comment- Reformat the numbering of the supplementary tables so that they appear in ascending order in the text and not randon (example, Table S7 is mentioned first in the text than Table S3).

Response 9: We are very sorry for our incorrect writing. We have re-written this part according to the Reviewer’s suggestion. Please see the manuscript in attachment.

Point 10: Table S5- Comment- As it is this table is totally redundant with the text. Instead of the numbers of genes containing those elements please write  the accession number of the genes that have each of the elements.

Response 10: We are very sorry for our incorrect writing. We have re-written this table according to the Reviewer’s suggestion. Please see the table in attachment.

Point 11: Table S6- Comment- What is the comparison base line in this data, what are these numbers referenced to? Please specify in a clear way.

Response 11: We are very sorry for our incorrect writing. We have re-written this part according to the Reviewer’s suggestion. “The cotton CHI gene FPKMs downloaded from ccNET and CottonFGD.” was added in table S6.

Point 12: Figure 7- Comment- This figure should be made bigger and only the genes that show different expression between the two cultivars should be shown. The others move to supplementary data.

Response 12: We are very sorry for our incorrect displaying. We have revised  Figure 7 and added Figure S1 according to the Reviewer’s suggestion. Please see the Figure 7 and Figure S1 in attachment.

Point 13: Figure 7-Please provide RT-PCR gel showing the induction of expression for the most interesting genes and at the most interesting data-points

Response 13: We are very sorry for our incorrect displaying. We have revised  Figure 7 and added RT-PCR gel showing the induction of expression for the most interesting genes and at the most interesting data-points according to the Reviewer’s suggestion. Please see the Figure 7  in attachment.

    We tried our best to improve the manuscript and made some changes in the manuscript.These changes will not influence the content and framewort of the paper. And here we did not list some changes but marked in red in revised paper.We appreciate for Editors/Reviewers’ warm work earnestly, and hope that the correction will meet with approval.

Once again, thank you very much for your comments and suggestions.

Reviewer 2 Report

I accept in current form

Author Response

Dear Editors and Reviewers:

    Thank you for your letter and for the reviewers’ comments concerning our manuscript entitled “The Chalcone Isomerase Family in Cotton: Whole-Genome Bioinformatic and Expression Analyses of the Gossypium barbadense L. Response to Fusarium Wilt Infection” (ID: genes-642085).  And our manuscript have underwent extensive English editing by American Journal Experts on November 1, 2019.

    Once again, thank you very much for your comments.

Reviewer 3 Report

Zu and colleagues performed a comparative bioinformatic study of chalcone isomerase (CHI) genes in four cotton species. Despite CHI representing an important enzyme in flavonoid metabolism, the CHI gene family has not been well characterized in cotton to date, making this study relevant. In addition, the authors investigated CHI gene expression upon infection with the phytopathogenic fungus Fusarium oxysporum and discuss a potential role of these genes for pathogen resistance.

The manuscript is well written and the methods used are described in sufficient detail. The bioinformatic analysis has been performed well and provides detailed insights relevant for understanding the CHI gene family, as well as phylogenetic relationships, gene conservation and evolution.

The Fusarium infection experiment reveals some interesting gene expression patterns. However, some points need to be addressed to support the statements in the manuscript.

How was the selection of the 10 candidate genes analyzed performed? Statistical analysis of the data in figure 7 is missing. ANOVA analysis would be appropriate to test the significance of differences as described in the main text. The legend for figure 7 is incomplete. Details on pathogen infection, replicate number, error bars etc. should be added. Replace “h” with “hpi”. The authors state that the two cotton cultivars “exhibited susceptibility and resistance to Fusarium wilt race 7, respectively” (p.12) and discuss that transcripts of 5 CHI genes “made a significant contribution to the defense against Fusarium wilt race 7” (p.14). The authors do not show any data on pathogen resistance/susceptibility to support their claims. Furthermore, causality between gene expression and pathogen defense cannot be deduced from the available data, but would require investigation of mutant and/or overexpression lines. On p. 14, authors state that “expression peaks at two different stages” have been identified. However, no pathogen interaction stages have been defined in the manuscript. Authors should show histological data of Fusarium growth to support this interpretation, or rephrase.

Author Response

Dear Editors and Reviewers:

    Thank you for your letter and for the reviewers’ comments concerning our manuscript entitled “The Chalcone Isomerase Family in Cotton: Whole-Genome Bioinformatic and Expression Analyses of the Gossypium barbadense L. Response to Fusarium Wilt Infection” (ID: genes-642085). Those comments are all valuable and very helpful for revising and improving our paper, as well as the important guiding significance to our researches. We have studied comments carefully and have made correction which we hope meet with approval. And our manuscript have underwent extensive English editing by American Journal Experts on November 1, 2019. Revised portion are marked in red in the paper. The main corrections in the paper and the responds to the reviewer’s comments are flowing:

Response to Reviewer 3 Comments

Point 1: How was the selection of the 10 candidate genes analyzed performed?

Response 1: We are very sorry for our unclear  writing. We have re-written this part according to the Reviewer’s suggestion. We have added “Our previous transcriptomes, which were obtained from the parents and the F6 RILs (two resistant offspring and two susceptible offspring) after Fusarium wilt infection, have shown that CHI genes have important function in cotton.” in our manuscript (p.3).

Point 2: Statistical analysis of the data in figure 7 is missing. ANOVA analysis would be appropriate to test the significance of differences as described in the main text. The legend for figure 7 is incomplete. Details on pathogen infection, replicate number, error bars etc. should be added. Replace “h” with “hpi”.

Response 2: We are very sorry for our incorrect displaying. We have revised  Figure 7 and added Statistical analysis of the data according to the Reviewer’s suggestion. “Error bars indicate standard errors of three biological replicates. Statistically significant differences between Xinhai-14 and 06-146 were calculated based on an independent Student’s t-tests: * p < 0.05; ** p < 0.01.” was also added in Figure 7 and Figure S1.Please see the Figure 7 and Figure S1 in attachment.

Point 3: The authors state that the two cotton cultivars “exhibited susceptibility and resistance to Fusarium wilt race 7, respectively” (p.12) and discuss that transcripts of 5 CHI genes “made a significant contribution to the defense against Fusarium wilt race 7” (p.14). The authors do not show any data on pathogen resistance/susceptibility to support their claims. Furthermore, causality between gene expression and pathogen defense cannot be deduced from the available data, but would require investigation of mutant and/or overexpression lines.  

Response 3: We are very sorry for our unclear description. We have rephrased and added reference to describe our claims about  pathogen resistance/susceptibility according to the Reviewer’s suggestion. The text of “As shown in Figure 7, thoes CHI genes responded to inoculation treatment with expression peaks at two different stages. Regarding the differential expression over time, this work considered the response time to the inoculation treatment, since former studies had shown that preformed defense responses include thickening of the cuticle, synthesis of phenolic compounds, changes in the cell wall structure, accumulation of reactive oxygen species, release of phytoalexins and the hypersensitive response” were corrected as “As shown in Figure 7, thoes CHI genes might be involved in the defense against Fusarium Wilt. Former studies had shown that preformed defense responses include thickening of the cuticle, synthesis of phenolic compounds, changes in the cell wall structure, accumulation of reactive oxygen species, release of phytoalexins and the hypersensitive response [67], and the responding mechanism between gene expression and pathogen defense needs to be further explored” (p.15, p.16). Please see the manuscript in attachment.

Point 3: On p. 14, authors state that “expression peaks at two different stages” have been identified. However, no pathogen interaction stages have been defined in the manuscript. Authors should show histological data of Fusarium growth to support this interpretation, or rephrase. 

Response 3: We are very sorry for our unclear description. We have rephrased according to the Reviewer’s suggestion. The text of “For instance, transcripts of GbCHI01, GbCHI05, GbCHI06, GbCHI09 and GbCHI10 were abundantly induced by inoculation treatments and made a significant contribution to the denfese against Fusarium wilt race 7  (Figure 7)” were corrected as “For instance, transcripts of GbCHI01, GbCHI05, GbCHI06, GbCHI09 and GbCHI10 were abundantly induced by inoculation treatments and made a mainly contribution to the biotic stress response preliminarily (Figure 7 and Figure S1).” and “Regarding the differential expression over time, this work considered the response time to the inoculation treatment, since former studies had shown that preformed defense responses include thickening of the cuticle, synthesis of phenolic compounds, changes in the cell wall structure, accumulation of reactive oxygen species, release of phytoalexins and the hypersensitive response” were corrected as “Former studies had shown that preformed defense responses include thickening of the cuticle, synthesis of phenolic compounds, changes in the cell wall structure, accumulation of reactive oxygen species, release of phytoalexins and the hypersensitive response [67], and the responding mechanism between gene expression and pathogen defense needs to be further explored” (p.15). Please see the manuscript in attachment.

    We tried our best to improve the manuscript and made some changes in the manuscript.These changes will not influence the content and framewort of the paper. And here we did not list some changes but marked in red in revised paper.We appreciate for Editors/Reviewers’ warm work earnestly, and hope that the correction will meet with approval.

    Once again, thank you very much for your comments and suggestions.
